# Development of pH-Responsive Polypills via Semi-Solid Extrusion 3D Printing

**DOI:** 10.3390/bioengineering10040402

**Published:** 2023-03-24

**Authors:** Fan Wang, Ling Li, Xiaolong Zhu, Feng Chen, Xiaoxiao Han

**Affiliations:** National Engineering Research Center for High-Efficiency Grinding, College of Mechanical and Vehicle Engineering, Hunan University, Changsha 410082, China

**Keywords:** pH-responsive medicine, semi-solid extrusion, carboxymethyl chitosan, sodium alginate, personalised medicine

## Abstract

The low bioavailability of orally administered drugs as a result of the instability in the gastrointestinal tract environment creates significant challenges to developing site-targeted drug delivery systems. This study proposes a novel hydrogel drug carrier using pH-responsive materials assisted with semi-solid extrusion 3D printing technology, enabling site-targeted drug release and customisation of temporal release profiles. The effects of material parameters on the pH-responsive behaviours of printed tablets were analysed thoroughly by investigating the swelling properties under both artificial gastric and intestinal fluids. It has been shown that high swelling rates at either acidic or alkaline conditions can be achieved by adjusting the mass ratio between sodium alginate and carboxymethyl chitosan, enabling site-targeted release. The drug release experiments reveal that gastric drug release can be achieved with a mass ratio of 1:3, whilst a ratio of 3:1 allows for intestinal release. Furthermore, controlled release is realised by tuning the infill density of the printing process. The method proposed in this study can not only significantly improve the bioavailability of oral drugs, but also offer the potential that each component of a compound drug tablet can be released in a controlled manner at a target location.

## 1. Introduction

Oral compounded tablet delivery is popular among various drug delivery systems because of its ease of administration and low patient pain [1]; it has also been adopted by the World Health Organization (WHO) as a strategy for improving patient compliance [2,3]. Patients with multiple underlying conditions (suffering from peptic ulcers, diabetes, and chronic diseases simultaneously, for instance) require multiple medications with sustained effects at different sites to reduce side effects and improve the efficiency of drugs. Therapeutic drugs are needed for sustained and controlled release in the stomach for gastric disorders such as peptic ulcers, while antineoplastic therapies for systemic diseases such as diabetes and chronic diseases usually require a high absorption of proteins and peptides in the intestine. Oral proteins and peptides can be broken by gastric juice while passing through the stomach due to the pH gradient in the gastrointestinal tract of humans, resulting in the low bioavailability of intestinal drugs [4]. Therefore, achieving the location-targeted release of compound drugs through the oral routine can improve the bioavailability of drugs.

pH-responsive release strategies assisted with 3D printing technology have been widely used for addressing the problems above. Of the many 3D printable biomaterials, pH-responsive hydrogels are preferred due to their high volume response to environmental pH changes. They can be utilised to regulate drug release at a specific location within human bodies. Sodium alginate (SA), one of the most widely used natural anionic biopolymers, is sensitive to pH values [5,6]. It can crosslink with divalent cations such as Ca^2+^, Ba^2+^, or Zn^2+^ to form three-dimensional networks [7,8,9]. A large number of carboxyl groups in SA hydrogels can become ionised under acidic conditions, leading to the shrinkage of structure that can impede the release of the loaded drug. In contrast, deprotonation occurring under alkaline conditions can expand the networks, thereby accelerating drug release [10]. Nonetheless, SA hydrogels possess weak mechanical properties and high degradation rates, leading to abrupt release [6,11]. It has been shown that such disadvantages of SA can be overcome by mixing amphoteric polyelectrolyte water-soluble carboxymethyl chitosan (CMCS) with SA [12,13]. Chitosan hydrogels containing SA have been widely adopted for sustained drug delivery in the stomach because of their high pH response under acidic conditions [14,15,16,17,18]. However, it is still challenging to develop a single pH-responsive drug delivery system that can be used for regulating both gastric and intestinal drug release.

3D printing technology is adopted broadly to fabricate personalised drug delivery systems due to its geometrical flexibility and multi-material printing feature, enabling drugs to be delivered over tailored release profiles [19,20,21]. Semi-solid extrusion (SSE), a material extrusion 3D printing method, has been adopted for manufacturing personalised drug delivery systems due to its low printing temperature and high drug loading capacity [22]; however, the printability of the material remains an issue, and high printing accuracy is challenging to be achieved [23]. Falcon et al. [23] improved the hydrogel printing performance by pre-crosslinking calcium ions in hydrogels. Wei et al. [24] used a mild semi-solid paste extrusion 3D printing technique to fabricate oral tablets with two different release kinetics using caffeine and vitamin B analogues as model drugs; however, the printed tablets undergo poly-stack shrinkage after drying, leading to low printing fidelity. The rheological properties of the semi-solids can also affect the printing accuracy and the geometry of the printed parts. Therefore, a new formula for drug release systems offering predominant printability and tunable release behaviours is urgently required.

In this study, pH-responsive tablets were fabricated using a new printing ink containing CMCS, SA, polyethylene glycol diacrylate (PEGDA), diphenyl(2,4,6-trimethylbenzoyl)phosphine oxide (TPO), and drugs. PEGDA, a pharmaceutical photocurable polymer, can be cured under UV light irradiation after the addition of photoinitiators TPO to improve the forming ability of semi-solid extrusion 3D printing. After light curing, the tablets were then crosslinked secondary with Ca^2+^ to enable the responsive capability to pH values. The impact of material parameters on pH-responsive behaviours was then thoroughly analysed. It has been found that location-dependent drug release can be achieved by adjusting the ratio between SA and CMCS, allowing for independent drug releases in the stomach or intestine. Furthermore, the rheological analysis of the printing ink was carried out, demonstrating strong shear-thinning behaviours and good recovery capability. Finally, controlled/sustained release was achieved by tuning the release profiles regulated by the infill density of different printing layers.

## 2. Materials and Methods

### 2.1. Materials

Sodium alginate (AR, 90%), carboxymethyl chitosan (BR, water-soluble), phenylbis (2.4.6-trimethyl benzoyl) phosphine oxide (MW = 418.16 g/mol, 98%), and PEGDA 575 (MW = 170.16 g/mol) were purchased from Shanghai Maclean Biochemical Technology Co., Ltd. (Shanghai, China) Bovine serum albumin (biotech grade, 96%) was obtained from Shanghai Aladdin Reagent Co., (Shanghai, China).

### 2.2. Preparation of Printing Ink

The photo-crosslinkable printing ink, composed of photoinitiator (TPO), photo crosslinker (PEGDA), drug (bovine serum protein), CMCS, SA, and water, was prepared by first adding TPO to PEGDA under the light-proof condition. The bovine serum protein powder was mixed with hyperpure water and stirred at room temperature until completely dissolved. The resulting bovine serum protein solution was then added to the aqueous PEGDA solution. Afterwards, the CMCS powder and SA were dissolved in the mixture developed above through magnetic stirring to form a printing ink. Finally, the obtained hydrogel paste was loaded into a 10 mL conical centrifuge tube and centrifuged at 3000 rpm for 15 min to remove air bubbles. The defoamed hydrogel paste was stored in a refrigerator (4 °C) for 16 h to eliminate the bubbles of the printing ink, improving the spatial uniformity of the printing ink, and therefore, the printability. The detailed composition of each sample of the printing ink used in this study was summarised in Table 1.

Different amounts of calcium chloride were accurately weighed and then dissolved in 50 mL of water to obtain the crosslinkable calcium ion solution (1% *w*/*v*, 2% *w*/*v*, 5% *w*/*v*, and 10% *w*/*v*) that can crosslink with SA. The resulting crosslinked SA networks enable the pH-responsive functionality of the printed tablets.

### 2.3. Printing of Tablets

The printing ink prepared previously was transferred to the conical syringe (0.26 mm orifice) of a semi-solid extrusion 3D printer (Regemat3D, Granada, Spain, see Figure 1a). The geometric of the tablets and printing parameters were set through the software (Regemat3D Designer) as follows: length: 12 mm; width: 12 mm; height: 4 mm; filling mode: diagonal; flow rate: 1.2 mm/s. The tablets were fabricated with various infill densities, including 50%, 75%, and 100% (see Figure 1b–d). A UV light source (12,000 mW/cm^2^) was loaded adjacent to the nozzle of the 3D printer and the tablets were fabricated in a layer-by-layer fashion. Specifically, the bioink with no light irradiation was extruded to form an uncrosslinked layer, followed by light irradiation along the extrusion path to cure the entire layer. The 3D tablets were then obtained by repeating such a layer-wise process.

### 2.4. Post-Treatment of Tablets

The printed hydrogel tablets were added to the calcium ion solution for secondary crosslinking. Residual calcium ions on the surface of tables were rinsed with ultrapure water. Subsequently, the deionised tablets were dried with filter paper and placed in a freezer (2 h) for rapid freezing. The pre-frozen hydrogel tablets were then quickly transferred to the freeze dryer (Scientz, Ningbo, China), in which they were lyophilised for more than 20 h at a temperature of −80 °C and a vacuum degree of 1 Pa. The lyophilised hydrogel tablets were then sealed and stored for further use.

### 2.5. Characterisation of Hydrogel Properties

A small amount of hydrogel was loaded onto the plate of a rotational rheometer (Anton Paar, Graz, Austria) to conduct the rheological analysis at shear rates from 0.1 S^−1^ to 100 S^−1^ in the flow mode at room temperature (25 °C) [25]. The shear recovery test was performed at three stages: a low shear rate of 0.1 S^−1^ for 100 s followed by a high shear rate of 100 S^−1^ for 30 s and a low shear rate of 0.1 S^−1^ for 100 s. The hydrogel was exposed to UV light for 0–600 s. The storage modulus and loss modulus were measured in the oscillatory mode (strain = 1% and angular frequency = 6.28 rad/s) to obtain frequency-loss modulus and frequency-storage modulus curves [26].

The microscopic morphology of the hydrogel was investigated under scanning electron microscopy (Tescan, Brno, Czech Republic) at 5 keV. Lyophilised hydrogel samples were cut with a thin blade and sprayed with gold.

The swelling rate (SR) was calculated according to the following formula [27]:(1)$${\mathrm{SR}\;(\%)=(W}_{t}-W_{0}{)/W}_{0}\;\times \;100\%$$
where $W_{0}$ is the initial dry weight and $W_{t}$ is the wet weight of the hydrogel tablet after swelling. The dried lyophilised hydrogel tablets were weighed initially and immersed in the hydrochloric acid solution (pH = 1.2) and PBS solution (pH = 7.4). The tablets were removed from the solution when fully swollen, and the residual water on the surface was removed with filter paper. After that, the tablets were weighed on an analytical balance to record their wet weight.

### 2.6. Preparation of Artificial Gastric and Intestinal Fluids

#### 2.6.1. Preparation of Hydrochloric Acid Solution with a pH Value of 1.2 (Artificial Gastric Fluid)

In total, 1 g of sodium chloride was dissolved in 400 mL of deionised water and 0.90 mL concentrated hydrochloric acid was added to a solution composed of 1 g sodium chloride and 400 mL deionised water, followed by magnetic stirring. The pH value of the solution was monitored using a PHS-3E (Leici, Shanghai, China) digital display acidity meter and adjusted to approximately 1.2 using 1.0 mol/L HCl to mimic the gastric fluid. The acidic solution was transferred to a 3000 mL volumetric flask for further use.

#### 2.6.2. Preparation of PBS Solution (Artificial Intestinal Solution) with a pH Value of 7.4

In total, 8 g NaCl, 1.44 g disodium hydrogen phosphate, 0.2 g potassium chloride, and 0.21 g sodium dihydrogen phosphate were dissolved in 800 mL deionised water. The pH value of the solution was tuned to be 7.4 with 1.0 mol/L NaOH, and the pH was monitored using a PHS-3E digital acidity meter. The alkaline solution was transferred to a 3000 mL volumetric flask for further use.

### 2.7. Fourier Transform Infrared Spectroscopy (FTIR)

Infrared spectra of all samples were acquired at room temperature using a Perkin-Elmer FTIR spectrometer (Perkin-Elmer, Waltham, MA, USA). The samples were scanned from 4000 cm^−1^ to 500 cm^−1^ at a resolution of 2.0 cm^−1^.

### 2.8. Drug Release Testing

Bovine serum protein was chosen as the drug model, and the equation of the standard release curve was established by determining the UV absorption of bovine serum protein solution at different concentrations. Specifically, bovine serum protein (0.025 g) was mixed with 50 mL ultrapure water, and 0.1 mL, 0.2 mL, and 0.4 mL of such a solution were then mixed with 9.9 mL, 9.8 mL, and 9.6 mL of hydrochloric acid buffer (pH = 1.2) and PBS buffer (pH = 7.4), respectively, yielding 6 types of solution with various concentrations (5μg/mL, 10 μg/mL and 20 μg/mL). The resulting solution was scanned at a wavelength range of 190–600 nm. The maximum absorption for pH 1.2 occurred at a wavelength of 201 nm, while it was 202 nm for pH 7.4. The standard equation of UV absorbance of bovine serum protein was obtained as follows [28]:(2)$${\mathrm{Conc}}_{\mathrm{pH}1.2}=(30.541)\mathrm{Abs},\;r=0.999375$$
(3)$${\mathrm{Conc}}_{\mathrm{pH}7.4}=(37.485)\mathrm{Abs},\;r=0.999643$$

In vitro drug release testing was performed following the paddle method [29]. The test conditions were configured as follows: speed of 100 rpm; dissolution medium of hydrochloric acid solution (pH = 1.2, 1000 mL) and PBS solution (pH = 7.4); and dissolution temperature of 37 °C. In total, 10 mL samples were withdrawn from the tablet solution at various time intervals to measure the UV absorbance using a UV-Vis spectrophotometer (Youke, Shanghai, China). The cumulative drug release was calculated based on the standard release curve developed previously.

## 3. Results and Discussion

### 3.1. Effects of Calcium ion Concentration and Crosslinking Time on pH-Responsive Properties

The printed tablets were post-treated with calcium ions to allow for secondary crosslinking so as to endow them with pH-responsive properties. The effects of calcium ion concentration on pH-responsive behaviours were investigated by crosslinking the hydrogels with 1%, 2%, 5%, or 10% of calcium ions. To minimise the impact from other parameters, the ratio between SA and CMCS was chosen to be 2:1 and the crosslinking time was kept constant (1h). Since the drug release properties of hydrogels depend on their swelling behaviours [30], studying the swelling behaviour of hydrogels appears to be crucial for controlled drug release [31]. As shown in Figure 2a, the hydrogel’s swelling rate decreased rapidly with the increase of CaCl_2_ (particularly for pH 7.4), demonstrating a significant correlation with the concentration of calcium ions. The swelling rate was reduced by approximately 11 times for pH 7.4 when increasing the concentration of CaCl_2_ from 1% to 10%, whereas it was ~3x for pH 1.2. Interestingly, when the concentration of CaCl_2_ was greater than ~5%, the swelling rate of the hydrogel at pH 7.4 was less than that of pH 1.2. This may be attributed to the excessive chelation between calcium ions and CMCS under an acidic condition, which can form new crosslinked networks, and therefore, enhance the swelling [32]. The ratio of the swelling rate that reflects the strength of pH response dropped when calcium ions were increased (see Figure 2b). High pH responses were observed when the CaCl_2_ concentration was low (1% and 2%), while it was suppressed remarkably when the CaCl_2_ concentration became high (5% and 10%). This was likely due to the increased crosslinkers that lessen the number of lattices between the molecular chains and affect the inward penetration of water molecules, thus lowering the swelling rate of hydrogels.

Figure 3 shows the optical morphology of the printed hydrogel tablets crosslinked with CaCl_2_ (8 h) at pH 1.2 and pH 7.4. As seen in Figure 3a–d, the size of the tablets decreased with the increase of CaCl_2_ concentration for both acidic and alkaline conditions. However, the swelling rate of the tablets at pH 7.4 was more prominent than that of pH 1.2. The swelling behaviours observed from the morphological images were consistent with that shown in Figure 2, demonstrating the potential that the pH-responsive characteristics of the drug-loaded hydrogels can be modulated by tuning the concentration of calcium ions.

To understand how the crosslinking time influences the pH-responsive behaviours of hydrogels, the printed tablets were crosslinked with a constant concentration of CaCl_2_ (2%) for various periods (0.25, 0.5, 1, 2, and 3 h), while the other parameters were identical. As shown in Figure 4a, the swelling of tablets with pH 7.4 reduced rapidly with increased crosslinking time, exhibiting strong sensitivity to crosslinking time. In contrast, the swelling rate was less affected for tablets with pH 1.2. Additionally, the swelling ratio (see Figure 4b) could be as large as ~3x when the crosslinking time was short (0.25 h, for instance), offering the potential to control drug release at different locations in human bodies. However, the discrepancy in swelling between pH 1.2 and pH 7.4 declined significantly when the CaCl_2_ crosslinking time was extended, indicating that pH values have a limited impact on swelling as long as the crosslinking time is long enough.

### 3.2. Effects of CMCS and PEGDA Concentrations on pH-Responsive Behaviours

The impacts of CMCS and PEGDA concentrations on pH-responsive behaviours were investigated by experimentally evaluating the sensitivities of the swelling rate of tablets. According to the previous results of this study, the concentration of CaCl_2_ and crosslinking time were selected to be 2% and 0.5 h, respectively, with which strong pH responses can occur. As shown in Figure 5a, the pH sensitivity of hydrogel containing only SA was higher than that of other hydrogels, indicating that both CMCS and PEGDA can deteriorate the pH responses of tablets. The swelling rate at pH 1.2 increased slightly by adding CMCS or PEGDA, whereas it decreased more rapidly under pH 7.4. This may be due to the protonation of amino groups in CMCS at pH 1.2, which can cause the generation of a large amount of free positive ions and electrostatic repulsion, therefore causing the crosslinking network to expand. Conversely, adding CMCS can enhance the electrostatic interactions among large molecules at pH 7.4, increasing the crosslinking density of hydrogels. As a result, the entry of water molecules into the hydrogel network is slower, thus hindering the swelling process [31]. Since the crosslinking between SA and CMCS forms a physical network, the chemical crosslinking of PEGDA is irreversible, thereby decreasing the pH response of the hydrogels. In addition, CMCS can lower the swelling rate of the hydrogel at pH 7.4 due to the presence of amino groups. To further understand the influence of PEGDA on pH response, the sensitivity analysis of swelling rate on the concentration of PEGDA was conducted (see Figure 5b). The pH sensitivity was enhanced as the concentration of PEGDA decreased, particularly for concentrations lower than 10%.

### 3.3. Effects of the Ratio between SA to CMCS on pH-Responsive Behaviours

To further understand how CMCS regulates the pH-responsive behaviours of hydrogels, a more quantitative analysis of the sensitivity of swelling rate to the ratio between SA and CMCS was carried out. Based on previous results in this study, other parameters, including the concentration of PEGDA, CaCl_2_ and the crosslinking time, were selected to be 5%, 2%. and 0.5 h, respectively. It can be observed that the swelling rate at pH 1.2 grew gradually with the increase of CMCS (see Figure 6a). The amino groups of CMCS that can be protonated enhance the swelling of the hydrogel network, leading to a higher swelling rate under acidic conditions. At pH 7.4, when a small portion of CMCS was added, the swelling rate of the hydrogel decreased. However, as the amount of CMCS increased, the hydrogel’s swelling rate gradually increased. When a higher amount of CMCS was added, the molecular chains were more tightly arranged, resulting in a greater intermolecular force of the hydrogel, and therefore, a higher crosslink density [33]. This prevents water molecules from entering the gel, impeding the swelling process. Increasing the number of CMCS produces numerous carboxyl groups that can undergo protonation under weak alkaline conditions, contributing to the swelling of the network. Figure 6b shows the pH response of aqueous hydrogel containing SA/CMCS with mass ratios less than 1. A ratio of 1:2 appeared to be a balance point, where the swelling rates of pH 1.2 and 7.4 were almost identical. When the ratio was less than 1:2 (the proportion of CMCS increased), the swelling rate of the hydrogel at pH 1.2 was higher than that of pH 7.4 due to excessive amino groups in CMCS. However, the swelling rate of pH 7.4 was significantly greater when the proportion of SA became dominant (see Figure 6a). Figure 6c depicts the swelling ratio between pH 7.4 and pH 1.2 against various mass ratios of SA/CMCS. The values of SR_pH7.4_:SR_pH1.2_ declined remarkably from 3.5 to 0.77 when the ratio between SA and CMCS was adjusted from 4:0 to 0:5, indicating that targeted drug release occurred in either the stomach or the intestine and can be achieved by tuning the ratio of CMCS to SA.

The tunability of the pH-responsive behaviours was further verified by examining the SEM microstructures of crosslinked hydrogels under acid and alkali conditions. Hydrogels containing SA and CMCS were immersed in both PBS (pH 7.4) and hydrochloric acid solution (pH 1.2) for swelling. The ratios of 1:3 and 3:1 were selected due to their high pH-responsive capability revealed by previous results. As observed in Figure 7a,b, the pores of the hydrogel (SA: CMCS = 3:1) collapsed in the hydrochloric acid solution (pH 1.2). In contrast, the hydrogel dissolved in PBS solution (pH 7.4) contained more porous structures and their pore walls were thinner than those in the hydrochloric acid solution, therefore possessing greater water absorption and swelling capacity. When the ratio between SA and CMCS was changed to 1:3 (Figure 7c,d), the pore walls at pH 1.2 were actually thinner than those at pH 7.4.

### 3.4. Fourier Transform Infrared Spectroscopy (FTIR)

The FTIR spectra of BSA, CMCS, SA, Prtintlet, and PEGDA are shown in Figure 8. The spectrum of PEGDA showed the characteristic peak of -CH_3_ at 2868 cm^−1^, a C=O stretching vibration peak at 1720 cm^−1^ and two C=C characteristic peaks at 810 cm^−1^ and 1636 cm^−1^. In the FTIR spectrum of Printlet, the stretching vibration peaks of the C=C bond at 810 cm^−1^ and 1636 cm^−1^ disappeared, indicating that the C=C bond underwent a photopolymerisation reaction and was converted to C-C bond [34]. The absorption peak of CMCS at 2920 cm^−1^ was due to the stretching vibration of -CH_2,_ and a peak at 1579 cm^−1^ was the characteristic absorption peak of -NH_2_. Additionally, a peak at 1402 cm^−1^ could be attributed to the symmetric stretching vibration of the -COO^−^ group, and a peak at 1049 cm^−1^ was the stretching vibration absorption peak of C-O. For SA, the absorption peak at 3266 cm^−1^ could be assigned to the stretching vibration of -OH; the absorption peak at 2922 cm^−1^ was due to the stretching vibration of -CH_2_; the absorption peaks at 1598 cm^−1^ and 1408 cm^−1^ were asymmetric and symmetric stretching vibration peaks of -COO^−^ group, respectively. For Printlet, the absorption peaks of SA at 1598 cm^−1^ and 1408 cm^−1^ and those of CMCS at 1579 cm^−1^ and 1402 cm^−1^ were shifted to 1592 cm^−1^ and 1412 cm^−1^, respectively. The shifts were caused by the interaction between SA and CMCS, suggesting that crosslinking occurred between SA and CMCS. Therefore, the networks formed in the prepared hydrogel tablets were composed of multiple interpenetrating networks, including the PEGDA chemically crosslinked networks and the SA/CMCS physically crosslinked networks.

### 3.5. Rheological Properties

In semi-solid extrusion 3D printing, the rheological properties of the printing ink can significantly influence the printing process and the formation of the structure [35,36]. The hydrogel ink in the syringe is initially in a flow-free state (0 shear rates). Subsequently, the plunger exerts a force on the hydrogel to provide a higher shear rate, extruding the ink from the nozzle onto the printing platform. The hydrogel eventually forms a predesigned shape and returns to its flow-free state. The key rheological properties associated with the printing process include viscosity, viscoelastic shear modulus, viscosity recovery behaviour, and shear stress [37]. As shown in Figure 9a, the viscosity of the hydrogel decreased rapidly when the shear stress was promoted from 0 to 800 Pa, demonstrating strong shear-thinning behaviour. As depicted in Figure 9b, the viscosity was lowered remarkably at a light exposure time of 120 s and rebounded by approximately two-thirds at 180 s, exhibiting excellent recovery capability. According to Figure 9c, both the energy storage modulus (G″) increased more rapidly with the increase of light exposure time than the loss modulus (G′), leading to the rapid descending of tanδ (see Figure 9d). Consequently, the hydrogel developed can form an elastic-dominant gel-like structure with good self-supporting performance.

### 3.6. In Vitro Drug Release

Figure 10a–c showed the drug release profiles of 3D-printed hydrogel tablets containing 5% PEGDA, 10% BSA, and SA/CMCS (3:1). Similar release patterns were observed under freeze drying, thermostatic drying and non-drying conditions, demonstrating a slight correlation to environmental conditions. The release rates were promoted by ~10% in a non-drying condition compared to the dry conditions, likely due to the contraction and collapse of the Printlet network structure during drying that can inhibit the diffusion of BSA [38]. In addition, it can be seen that the drug release rate of freeze-dried hydrogels was somewhat faster compared to that of thermostatically dried hydrogels. Water within the network structure of freeze-dried hydrogels can sublimate at low temperatures, and the original physical structure of hydrogels can be well maintained. Therefore, the pores of freeze-dried hydrogels are more sparse, resulting in larger internal porosity and specific surface area that conduce water absorption and thus improve swelling capacity [39].

Distinct pH responses were observed for all hydrogel tablets consisting of SA/CMCS (3:1), as displayed in Figure 10a–c. The sensitivity of the release rate to the pH values was less during the first 7.5 h, while the discrepancy between pH 1.2 and 7.4 was enlarged significantly after. The fractional release of drugs within 10 h was only ~10% at pH 1.2, indicating that the drugs were well protected in an acidic condition. This is due to the protonation of SA in the hydrochloric acid solution (pH 1.2) and the crosslinking between SA and CMCS through hydrogen bonding and electrostatic interactions, leading to the formation of dense networks of interpenetrating polymers that can prevent the outward diffusion of BSA [30]. Conversely, the fractional release can be as large as approximately 80–90% at pH 7.4, signifying that the drugs were released properly. The reason is that the deprotonation of carboxyl groups in PBS solution (pH 7.4) leads to the expansion of the hydrogel networks, therefore accelerating drug releasing. When the ratio of SA to CMCS was changed to 1:3 (see Figure 10d), the drug release rate under acidic conditions was greatly enhanced, which is suitable for stomach-targeted drug release. The amino groups in CMCS at pH 1.2 are able to be positively charged due to protonation, which can generate a repulsive force that promotes the swelling of the hydrogel networks. The results revealed that the location-dependent release of drugs (intestine or stomach, for example) can be achieved by adjusting the ratio between SA and CMCS.

According to the Noyes-Whitney equation [40]:(4)$$\frac{\mathrm{dc}}{\mathrm{dt}}{=k}_{D}A\;\left(C_{s}-C\right)\;$$
where dC/dt is the dissolution rate; k_D_ is the rate constant; A is the specific surface area; C_S_ is the saturation solubility of the drug; C is the concentration of the drug in the dissolution medium, the dissolution rate of drugs is proportional to the specific surface area. Hence, it is also possible to control the release rate by regulating the specific surface area of printed tablets, which can be achieved by changing the infill density of printing. In this study, tablets with three different infill densities (50%, 75%, and 100%) were manufactured using the semi-solid extrusion 3D printing method. As shown in Figure 11, the BSA release rate increased with the reduction of infill density, which is consistent with the results reported among other studies [41,42]. The release fraction was amplified by ~1.8x at 10 h when the infill density dropped from 100% to 50%.

## 4. Conclusions

This study presents an innovative drug delivery system that can modulate the release location and rate of drugs by combining semi-solid extrusion 3D printing technology with pH-responsive materials. A new printing ink consisting of CMCS, SA, PEGDA, TPO, and drugs was developed to fabricate 3D drug delivery systems, and the pH-responsive capability was enabled through secondary crosslinking with Ca^2+^. The pH-responsive behaviours of the printed tablets were then studied experimentally by investigating the impact of material parameters on swelling, exhibiting strong sensitivity to the concentration of Ca^2+^ and PEGDA, the mass ratio between SA and CMCS, and secondary crosslinking time. Distinct discrepancies in swelling at different pH values can be obtained by tuning the materials mentioned above, allowing site-targeted drug release, and optimal parameters (2% Ca2+, 5% PEGDA, and 30 min secondary crosslinking time) were also found. Additionally, specific release profiles can be achieved by adjusting the infill density of the printing, enabling controlled or sustained drug release. The method proposed in this study can not only significantly improve the bioavailability of oral drugs, patient compliance and therapeutic efficiency, but also offer the potential that each component of a compound drug tablet can be released in a controlled manner at a target location.

## Figures and Tables

**Figure 1 bioengineering-10-00402-f001:**
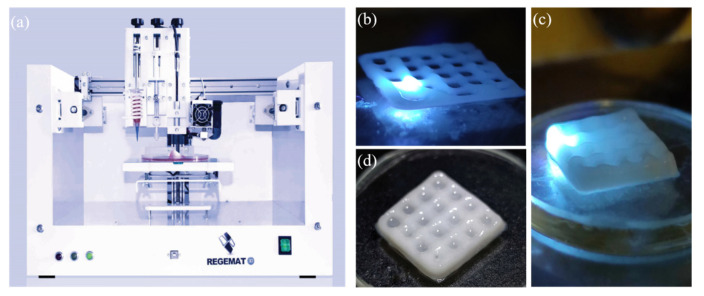
Semi-solid extrusion 3D printing. (**a**) Semi-solid extrusion 3D printer. Scaffolds printed with (**b**) 50% infill density, (**c**) 75% infill density, and (**d**) 100% infill density.

**Figure 2 bioengineering-10-00402-f002:**
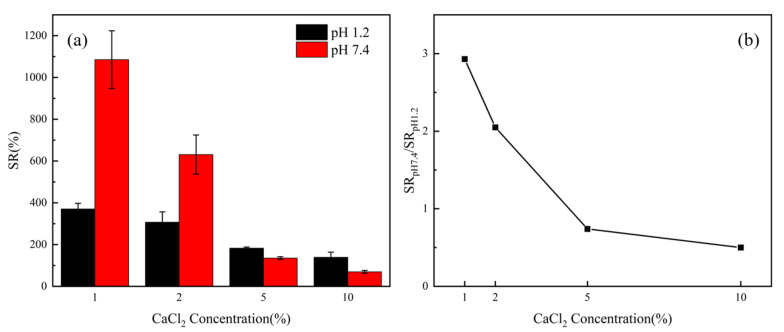
Effect of calcium ion concentration on pH-responsive properties of hydrogels. (**a**) Swelling rate of hydrogels crosslinked with CaCl_2_ at various concentrations for pH 1.2 and pH 7.4. (**b**) Ratio of the swelling rate of hydrogels at pH 1.2 and 7.4 as a function of calcium ion concentration.

**Figure 3 bioengineering-10-00402-f003:**
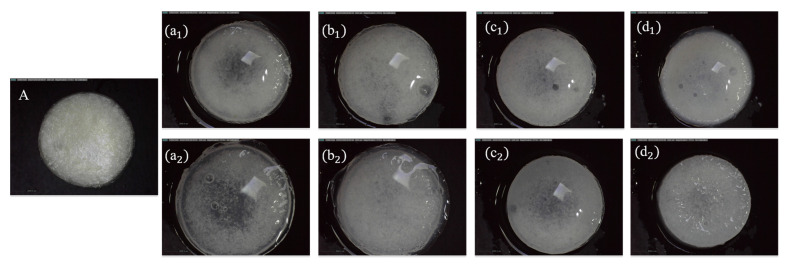
Swelling of hydrogels crosslinked with CaCl_2_ at different concentrations. (**A**) Unswollen hydrogel. Swollen hydrogels crosslinked with calcium at CaCl_2_ concentrations of 1%, 2%, 5%, and 10% for pH 1.2 (**a_1_**–**d_1_**) and pH 7.4 (**a_2_**–**d_2_**).

**Figure 4 bioengineering-10-00402-f004:**
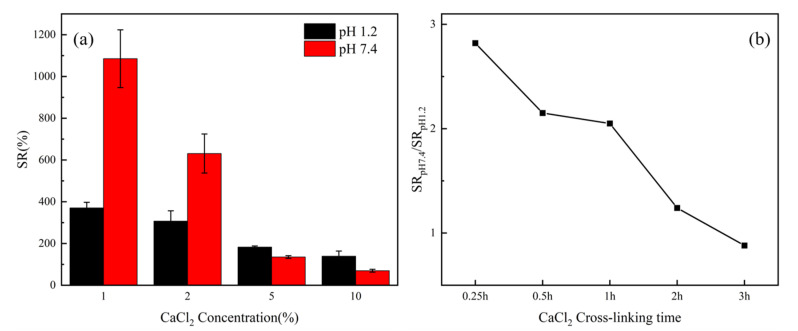
Effects of calcium ion crosslinking time on pH-responsive properties of hydrogels. (**a**) Swelling rate of hydrogels crosslinked with CaCl_2_ for different crosslinking times at pH 1.2 and pH 7.4. (**b**) Ratio of the swelling rate of hydrogels at pH 1.2 and pH 7.4 as a function of CaCl_2_ crosslinking time.

**Figure 5 bioengineering-10-00402-f005:**
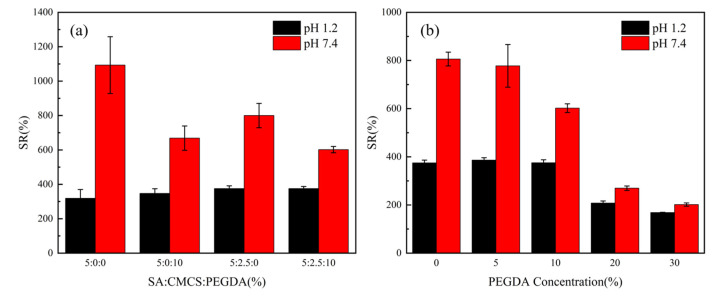
(**a**) Swelling rate of hydrogels containing SA, CMCS, and PEGDA that possess different proportions. (**b**) Swelling rate of hydrogels with various concentrations of PEGDA.

**Figure 6 bioengineering-10-00402-f006:**
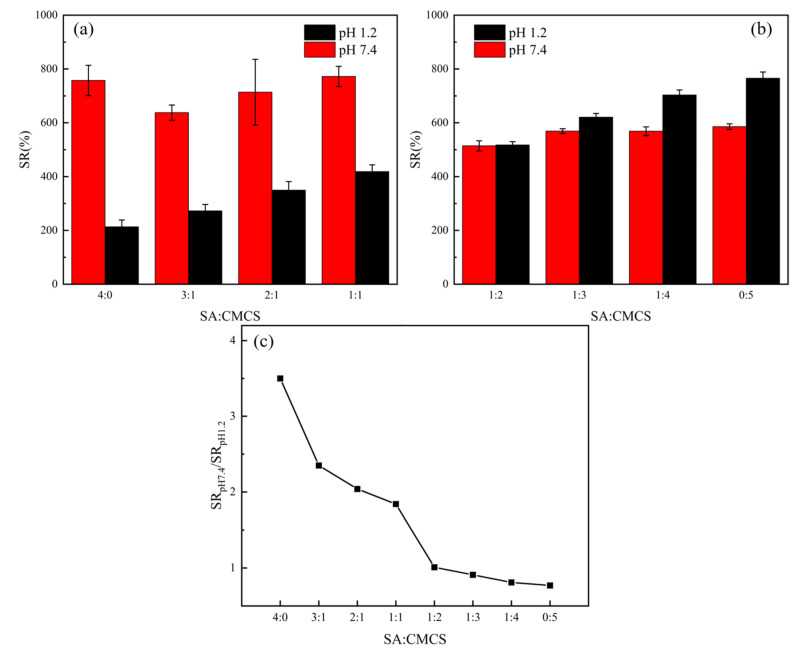
(**a**,**b**) Swelling rate between SA and CMCS on pH-responsive properties of hydrogels. (**c**) Ratio of the swelling rate between pH 7.4 and 1.2.

**Figure 7 bioengineering-10-00402-f007:**
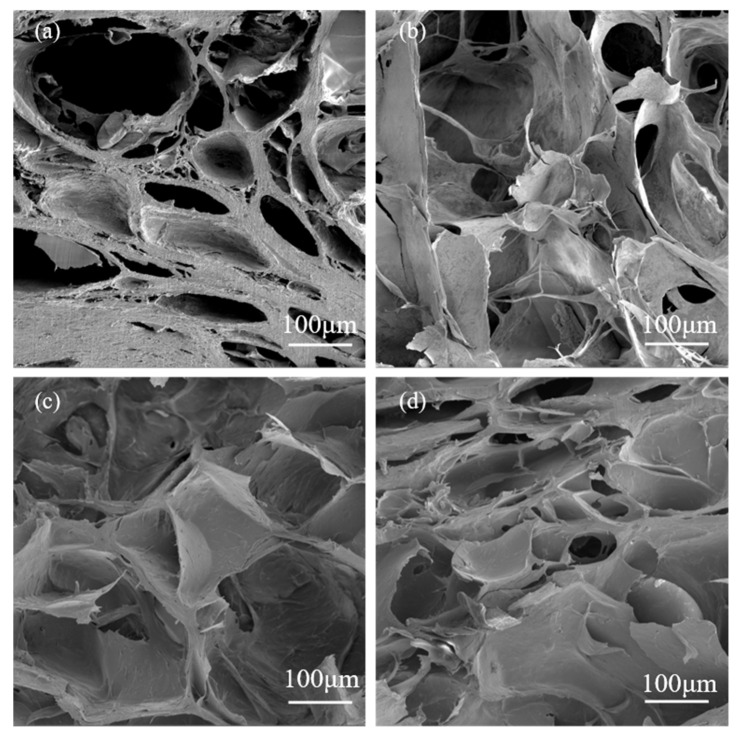
Cross-sectional SEM images of hydrogel after swelling in a solution at pH 1.2 (**a**,**c**) and pH 7.4 (**b**,**d**).

**Figure 8 bioengineering-10-00402-f008:**
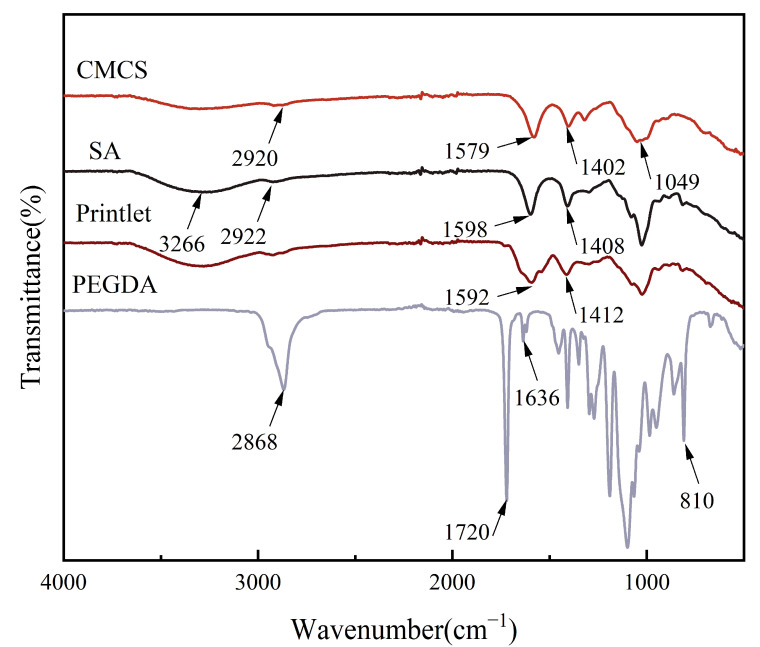
FTIR spectra of BSA, CMCS, SA, Printlet, and PEGDA.

**Figure 9 bioengineering-10-00402-f009:**
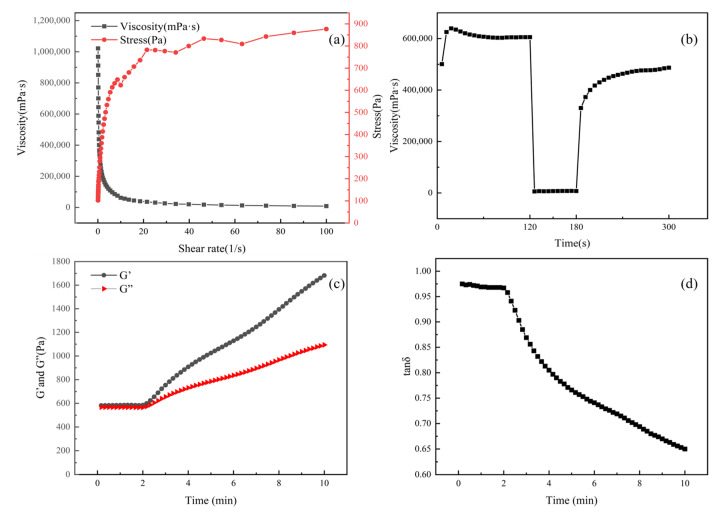
Rheological properties of hydrogels containing 5% PEGDA and SA/CMCS (3:1). (**a**) Viscosity versus shear rate. (**b**) Viscosity versus light exposure time. (**c**) Loss modulus and energy storage modulus. (**d**) Ratio between loss modulus and energy storage modulus.

**Figure 10 bioengineering-10-00402-f010:**
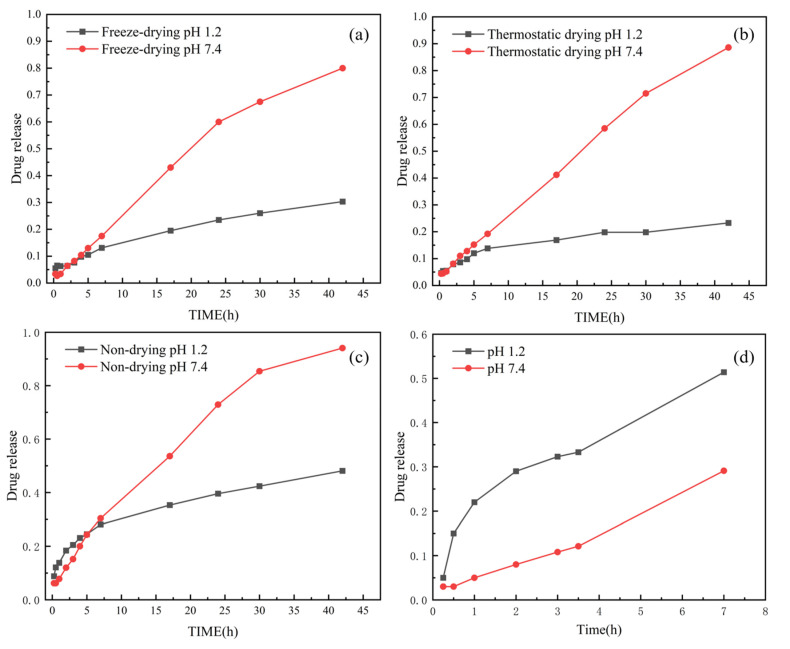
BSA release profiles under different conditions: (**a**) SA: CMCS = 3:1 and freeze-dried; (**b**) SA: CMCS = 3:1 and constant temperature-dried; (**c**) SA: CMCS = 3:1 and non-dried; and (**d**) SA: CMCS = 1:3.

**Figure 11 bioengineering-10-00402-f011:**
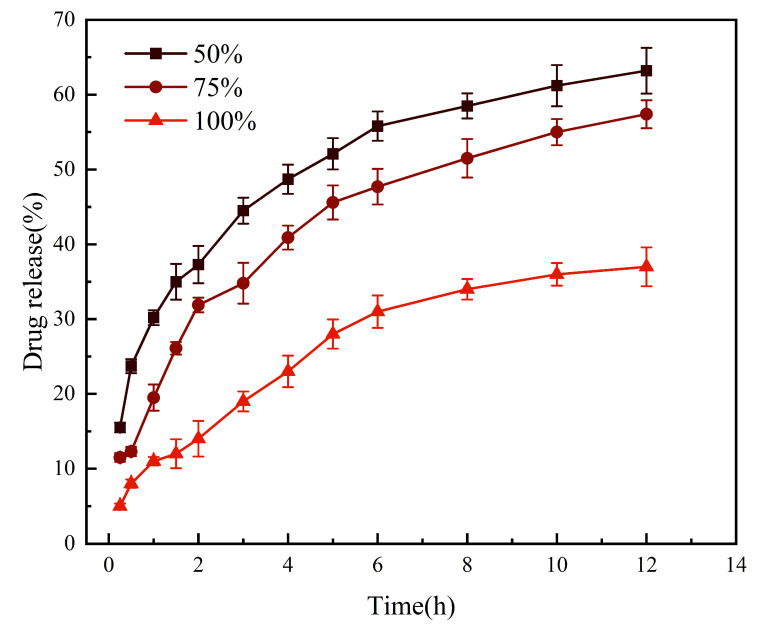
Drug release profiles of the tablets printed with various infill densities.

**Table 1 bioengineering-10-00402-t001:** Composition of the printing ink samples.

Sample	PEGDA/g	H_2_O/g	TPO/g	SA/g	CMCS/g	BSA/g	SA/CMCS
P1	0(0%)	20	0.1	1	0.5	2	2
P2	1(5%)	19	0.1	1	0.5	2	2
P3	2(10%)	18	0.1	1	0.5	2	2
P4	4(20%)	16	0.1	1	0.5	2	2
P5	6(30%)	14	0.1	1	0.5	2	2
W1	0	20	0	1	0	2	-
W2	2	18	0	1	0	2	-
W3	0	20	0	1	0.5	2	2
W4	2	18	0.1	1	0.5	2	2
B1	1	19	0.1	1	1	2	1
B2	1	19	0.1	1	0.5	2	2
B3	1	19	0.1	1	0.33	2	3
B4	1	19	0.1	1	0	2	-
B5	1	19	0.1	0	1	2	0
B6	1	19	0.1	0.25	1	2	0.25
B7	1	19	0.1	0.33	1	2	0.33
B8	1	19	0.1	0.5	1	2	0.5

## Data Availability

The data that support the findings of this study are available from the corresponding author upon reasonable request. All figures in this paper are original.

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
