# Peer review of "Development of pH-Responsive Polypills via Semi-Solid Extrusion 3D Printing"

_bioengineering, 2023, doi:10.3390/bioengineering10040402_

Round 1

Reviewer 1 Report

The present manuscript describes a way to fabricate drug delivery system through 3D printing. 

The manuscript lacks numerous details and clarifications before deserving publication.

1- The rationale behind the use of PEGDA plus alginate is not clearly explained. This lack of information is also obvious in the printing description part. Does the UV light is used during printing? What are the conditions?

2- In relation with this unclear strategy, results are presented where CaCl2 concentrations are varying but where conclusion are given about light-exposure time effect (Figure 4 and to of page 7 text). This is confusing and need to be refined

3- Page 9 and Figure 7 caption, the authors are talking for the first time of "dissolved", whereas everywhere else they mention "swelling". Again this is not clear enough to deserve publication.

Author Response

请参阅附件。

Reviewer 2 Report

1. Section 2.2, elaborate the line "enhance the homogeneity of the printing ink". What doe it mean.

2. How crosslinked SA behaves as a pH responsive material. Justify

3. Defend the novelty of the work.

4. How the author defend the scalability and commercialization on of the work.

 5. Figure 11 SD errorbar to be included in the figure.

6. Percent drug release graph to be provided.

Round 2

Reviewer 1 Report

Thank you to the authores for their corrections and clarifications.